# Contribution of Morphoanatomic Characters to the Taxonomy of the Genus LAELIA (Orchidaceae) in Mexico and Their Implication in Environmental Adaptation

**DOI:** 10.3390/plants12051089

**Published:** 2023-03-01

**Authors:** Montserrat Castellanos-Ramírez, Ulises Rosas, María Concepción Guzmán-Ramos, Estela Sandoval-Zapotitla

**Affiliations:** Jardín Botánico, Instituto de Biología, Universidad Nacional Autónoma de México, Mexico City 04510, Mexico

**Keywords:** plant morphoanatomy, Laeliinae, similarity analysis, altitudinal distribution, vegetative structural characters

## Abstract

The Laeliinae Subtribe (Epidendroideae-Orchidaceae) is exclusively Neotropical, composed of 43 genera and 1466 species, presenting great floral and vegetative diversity. The genus *Laelia* has restricted geographic distribution, with species in Brazil and Mexico. However, in molecular studies, the Brazilian species have not been included despite remarkably similar floral structures in both Mexican and Brazilian groups of species. The main objective of the present study is to analyze vegetative structural characters of 12 species of *Laelia* in Mexico to find similarities to recognize them as a taxonomic group and their relationship with possible ecological adaptations. This work supports the proposal to recognize 12 species of Laelias from Mexico as a taxonomic group (except for the new species recognized as *Laelia dawsonii* J. Anderson), since a strong similarity was found, translated by 90% of structural characters shared by the Mexican Laelias, given the relationship between the structural characters and the altitudinal distribution in which the groups of Mexican Laelias species are found. We propose that Laelias of Mexico be recognized as a taxonomic group and their structural characteristics allow for better understanding of adaptation of species to their environment.

## 1. Introduction

The Orchidaceae family is one of the most diverse plant groups, with around 30,000 species known worldwide [1,2]. Orchids are distributed on all continents (except Antarctica), but their greatest diversity is concentrated in tropical regions. Mexico, located on the northern limit of the American Tropics, is home to a remarkable wealth of orchids, and around 1260 species and 170 genera have been recorded in the country [3,4]. It is estimated that around 40% of Mexican orchids are endemic [5]. The number of taxa continues to increase due to the discovery of both species new to science and species from other countries not found in Mexico.

Anatomy, or the study of the shape and internal structure of plant tissues and organs, is a classic source of data used in plant taxonomy. These data are especially useful for phylogenetic relationships by suggesting homologies and contextualizing evolutionary polarity [6]. Currently it is recognized that anatomical characters can be as important as morphological ones. If strong similarities are found in the structural characters between Mexican species of *Laelia*, then it is proposed to recognize the Laelias of Mexico as a consolidated taxonomic group. The purpose of the present investigation is whether the Laelias of Mexico can be distinguished from the closely related genera *Broughtonia* and *Cattleya* based on structural characters.

To understand the similarity relationships between the Mexican species of *Laelia*, structural characters (morphological and anatomical) were analyzed for 15 Laeliinae species: 12 species of orchids that represent the genus *Laelia* in Mexico, *Broughtonia* (one species) and *Cattleya* (two species) that are genera closely related to *Laelia* [7], as comparison groups [8]. Furthermore, *Laelia dawsonii* has just been recognized as a distinct species from *L. anceps* [9] and was characterized as an endangered species in the most recent update of the list of threatened species in Mexico [10].

## 2. Results

### 2.1. Foliar Anatomical Description of the Laelia Group from Mexico

**Dermal Tissue**. Adaxial cuticle generally striated, except in *L*. *rubescens* and *L*. *halbingeriana*, which is smooth; adaxial cuticle thickness from 5 to 11 μm (<12 μm); except from *L*. *rubescens*, which is >12 μm, abaxial cuticle thickness 5–14 μm. Adaxial epidermal cells distributed in rows in *L*. *gouldiana*, *L*. *rubescens* and *L*. *halbingeriana*, the rest of the species with alternately distributed cells, the abaxial epidermal cells are in rows. Adaxial and abaxial epidermal cells are generally isodiametric polygonal in shape, except *L*. *eyermaniana*, *L*. *rubescens* and *L*. *superbiens*, where they are elongated polygonal. Area of adaxial epidermal cells <5000 μm^2^, cells abaxial from 801 to 1400 μm^2^, with the exception of *L. aurea*, where it is from 600 to 800 μm^2^. Stomata only in the abaxial epidermis, distributed in rows, except in *L*. *albida* and *L*. *autumnalis, L*. *furfuracea, L*. *superbiens* and *L*. *halbingeriana*, which is alternate, trichomes present at the margins of the adaxial surface, denoted by the permanence of basal cells, except in *L*. *albida, L. crawshayana, L*. *eyermaniana* and *L*. *halbingeriana*. Papillae present in the abaxial epidermis, except in *L*. *albida*, *L*. *eyermaniana*, *L*. *furfuracea* and *L*. *halbingeriana*, the papillae are epidermal, except in *L*. *anceps* and *L*. *speciosa*, where they are cuticular. The length of the guard cells is 30 to 40 μm, except in *L*. *superbiens*, which is larger than 45 μm. The development of the cuticular ridges is light in *L. albida, L. anceps, L. eyermaniana* and *L. halbingeriana* and deep in the rest of the species. The substomatal chamber is conspicuous, with the exception of *L*. *aurea, L*. *rubescens* and *L*. *halbingeriana*, which is inconspicuous. Stomata are generally semi-sunken, with the exception of that in *L*. *autumnalis, L*. *eyermaniana, L*. *furfuracea* and *L. speciosa* that are at the level of the epidermis and in *L. aurea* that are sunk. Stomatal index from 2 to 6. Height of adaxial epidermal cells 18–24 μm, except in *L*. *albida*, *L*. *anceps*, *L*. *autumnalis*, *L*. *furfuracea* and *L*. *gouldiana* ranging from 24 to 30 μm. Cell width adaxial epidermal 23–27 μm in *L*. *anceps* and *L*. *crawshayana,* 28–31 μm in *L. aurea* and in *L*. *furfuracea, L*. *speciosa* and *L*. *superbiens*, it Is 32–35 μm. Height of abaxial epidermal cells less than 30 μm as in *Laelia furfuracea*. Width of abaxial epidermal cells 25–37 μm as in *L. aurea,* except in *L*. *speciosa* which is 38 to 41 μm. **Fundamental tissue**. Hypodermis. Present in the leaf of all species, located both on the adaxial surface as abaxial, with the exception of *L. speciosa* which only occurs in the adaxial surface. Species with both hypodermises have a stratum in its abaxial hypodermis, the adaxial hypodermis generally presents one layer, with the exception of that in *L*. *superbiens*, which has two layers. Cell shape of the adaxial hypodermis is oblong except for *L*. *rubescens*, which presents cells with an elongated polygonal shape, and *L*. *speciosa* with polygonal cells isodiametric. Shape of cells of hypodermis abaxial, polygonal isodiametric except in the case of *L*. *anceps*, *L*. *superbiens* and *L*. *halbingeriana*, which are oblongs, and *L*. *Rubescens* that presents cells with a polygonal shape perpendicularly elongated. Thick cell walls in the adaxial hypodermis, except in *L. crawshayana* and *L*. *rubescens*, which are thin. In the hypodermis abaxial the walls are in some species thick and in others thin. The type of wall in both the adaxial and abaxial hypodermis is cellulosic, except in *L*. *superbiens* and *L*. *halbingeriana*, which is lignified. The distribution of the layers of the adaxial hypodermis is continuous for all species, while the distribution in the abaxial hypodermis is continuous only in some species, with the exception of that in *L*. *crawshayana*, *L*. *aurea*, *L*. *anceps*, *L*. *eyermaniana* and *L*. *speciosa*, which is discontinuous. On the other hand, in the basal region, the presence of hypodermis was found in all species. The location is on both surfaces (adaxial/abaxial), with the exception of *L*. *eyermaniana*, where it is abaxial only. There is one layer of the adaxial hypodermis, with the exception of *L*. *superbiens* and *L*. *crawshayana*, which have two or more layers. In the case of the abaxial hypodermis, there is only one layer for all species. Mesophyll with a thickness from 500 to 2500 μm, with *L*. *superbiens* being the species with the highest thickness (2500 μm). Heterogeneous mesophyll (palisade/spongy), except in *L*. *autumnalis, L*. *gouldiana, L*. *rubescens* and *L*. *superbiens*, which is homogeneous. The proportion of palisade parenchyma was generally 0 to 0.4, with the exception of that in *L*. *albida, L*. *anceps* and *L. aurea,* which is from 0.41 to 0.8, and in *L*. *rubescens, L*. *superbiens* and *L*. *halbingeriana*, which was 0.81 to 1. The presence of secondary thickenings was found only in the species *L*. *aurea* and *L*. *superbiens*. Water cells are absent in all species. Presence of calcium oxalate crystals with the exception of *L*. *albida*, *L*. *anceps, L*. *aurea*, *L*. *autumnalis* and *L*. *eyermaniana*. The shape of these crystals is mainly raphides. The ubication of the crystals is mainly in the center of the mesophyll. In the case of the basal region, the mesophyll is heterogeneous (spongy/palisade), with exception of the species that present a homogeneous mesophyll: *L*. *anceps*, *L*. *autumnalis*, *L*. *eyermaniana*, *L*. *gouldiana* and *L*. *rubescens*. The basal region also presents air spaces in the mesophyll, with the exception of *L*. *crawshayana*, *L*. *furfuracea*, *L*. *speciosa*, *L*. *superbiens and L. halbingeriana*. Extravascular fibers are found in the middle region in all species, the location is generally both abaxial and adaxial, with the exception of *L. rubescens* which has extravascular fibers present throughout the mesophyll. There are one to four layers of adaxial extravascular fiber bundles, with exception of *L*. *rubescens* and *L*. *superbiens*, which have four to five layers, make abaxial extravascular fibers of two to four layers, with the exception of *L*. *aurea and L*. *furfuracea*, which have four to six layers, and *L*. *rubescens*, which has six to nine layers. Extravascular fiber wall thickness of 2.5 μm, except for the species *L*. *aurea*, *L*. *gouldiana*, *L*. *superbiens* and *L*. *halbingeriana* with a wall thickness of 5 μm. Extravascular fibers with presence of stegmata. **Central zone**. In the central zone of the leaves, the presence of bulliform cells was found, with the exception of *L*. *anceps*, *L*. *aurea*, *L*. *eyermaniana*, *L*. *gouldiana* and *L*. *halbingeriana*. Species with bulliform cells have two to four strata. In a cross section, the leaves of the Mexican species of the genus *Laelia* present the projected central zone; this projection can be observed generally in convex shape, with the exception of that in *L*. *anceps*, *L*. *aurea*, *L*. *rubescens* and *L*. *superbiens*, which is keeled. Most species have a vascular bundle in the central zone. Regarding the presence of extravascular fiber bundles in the central zone, they were evident in all the species and generally had 2 to 4 strata. **Vascular tissue**. Four types of vascular bundles were found, except for *L*. *crawshayana, L*. *superbiens* and *L*. *halbingeriana* presenting three types and *L*. *rubescens* with five types of vascular bundles. In the blade, there are generally two layers of vascular bundles as in the case of *L*. *albida*, with the exception of *L*. *eyermaniana* and *L*. *superbiens* with one layer, and *L*. *rubescens* with three layers. Across the lamina there are 30 to 90 vascular bundles, with the exception of *L*. *superbiens* that presents more than 90 vascular bundles.

### 2.2. Evaluation of Quantitative Anatomical Characters of the Genus Laelia in Mexico

In particular, the characters evaluated were 11 quantitative anatomical characters with 30 repeated measurements per species. The summary of descriptive statistics for these characters, such as the mean (M) and the standard deviation (SD) are concentrated in Table 1. Each letter represents a group derived from the post hoc analysis.

### 2.3. Canonical Discriminant Analysis (CDA)

The CDA result showed that the first three canonical functions contributed 90% of the variation (Table 2). The eigenvalues and canonical correlation values were high (values greater than 1), which indicates that the evaluated parameters contributed greatly to the discrimination of groups and that the characters that displayed the greatest weight were the area of the abaxial epidermal cells, abaxial cuticle thickness and mesophyll thickness. On the other hand, some characters had values less than 1, in which we could observe that the height of the adaxial epidermal cells and the thickness of the adaxial cuticle could also be functional characters for the separation of the groups.

In the canonical discriminant analysis (CDA) (Figure 1) it was found that in canonical axis 1 (58.07%), the species *Laelia superbiens* and the genus *Cattleya* displayed a very thick mesophyll (1501 to 2000 µm), while on the left side were the *Laelia* species *L*. *crawshayana*, *L*. *eyermaniana*, *L*. *gouldiana* and *L. furfuracea*, with lower thickness of the mesophyll (500 to 1000 µm). This quantitative anatomical character showed variations within the Mexican Laelias species (Figure 2 and Figure 3).

In canonical axis 2 (25%), the two species of *Cattleya* had a smaller area of abaxial epidermal cells (600 to 800 µm^2^) compared to those of *L*. *crawshayana*, *L*. *eyermaniana*, *L*. *gouldiana*, *L*. *furfuracea*, *L*. *speciosa* and *L*. *anceps* (Figure 1) (800 to 1400 µm^2^; Figure 4 and Figure 5). Regarding the thickness of the abaxial cuticle, we could see at the lower end of the canonical axis 2 the genus *Cattleya* with the greatest thickness of the abaxial cuticle (≥18 µm), and at the upper end of the axis, the Mexican species of the genus *Laelia* with a smaller abaxial cuticle (≤18 µm; Figure 6 and Figure 7).

### 2.4. Cluster Analysis

The cluster analysis was obtained using the taxonomic distance coefficient and the average bond (UPGMA) as a grouping algorithm, separating the group of Laelias (pink) from the *Broughtonia* (green) and *Cattleya* species (red) (Figure 8). It was found that there was a similarity of structural characters in species that inhabited a certain altitude, particularly for the species of Mexican Laelias.

High-mountain Laelias (2000–3000 m asl), which include the species *Laelia anceps*, *L*. *albida*, *L*. *autumnalis*, *L*. *eyermaniana*, *L*. *furfuracea*, *L*. *gouldiana* and *L*. *speciosa*, share the following anatomical characters: striated cuticular surface, conspicuous substomatal chamber, thick cells walls of adaxial hypodermis, distribution of continuous adaxial hypodermis strata, absence of cells with secondary thickenings in the mesophyll and four types of vascular bundles in the mesophyll (based on the number of vascular bundles along the lamina) (Appendix A). The morphological characters shared by this group of Laelias are fleshy leathery leaves and sulcate pseudobulbs, glabrous lip, peduncle bracts much shorter than the internodes, sub-distichal arrangement of the flowers, lip throat without a brown spot, stigmatic surface hidden by rostellum and flower arrangement in raceme (Appendix A).

At the other extreme are the orchids at low altitudinal distribution (500 m asl), which include *Laelia aurea* and *L*. *rubescens* species. They are similar in the following anatomical characters: epidermal papillae present in the abaxial epidermis of leaves, deep development of cuticular edges of the stoma, inconspicuous substomatal chamber, cellulosic cell wall in the abaxial hypodermis of leaves, presence of secondary thickenings in the mesophyll and hypodermis of the basal region of a stratum and air spaces present in the basal region of the leaf. Interesting morphological characters that can be highlighted for lowland orchids are: chartaceous leathery leaves and pseudobulbs with a discoid shape and more wrinkled, a leaf in each pseudobulb, an exclusively pubescent lip, a lip throat with a brown spot, scale-shaped floral bracts, and a stigmatic surface that is not hidden by the rostellum.

Regarding the species of the analyzed genus *Cattleya*, in the case of *C. mossiae*, which is found at altitudes of 800 to 2000 meters asl, and for *C. purpurata* 500 meters asl, anatomical characters were identified that can be functional for distinguishing them from the genus *Laelia*, such as: stomatal index (>7), rough cuticle, oblong adaxial and abaxial epidermal cell shape. The morphological characters in both *Cattleya* species are leaves with proportions greater than 7.41 cm.

## 3. Discussion

Here we present a study where anatomical and morphological characters were integrated to analyze the similarity relationships and the altitudinal distribution present in the species of the genus *Laelia* in Mexico. The importance of the results obtained lies in the fact that the present research work shows that the Laelias of México share several structural attributes by which its species can be recognized as a taxonomic group.

Previous studies mention the relevance of the taxonomic relationships that exist between some species of the Laeliinae Subtribe based on their vegetative and floral morphology [11]. Stern and Carlsward [12], pointed out that exhaustive anatomical studies in the Laeliinae subtribe are scarce and that plant anatomy is particularly important to understand the relationships of the species in conjunction with their morphology. Baker characterized in detail the morphology and foliar anatomy of 36 genera and 79 taxa of the Laeliinae [13]. However, he did not include all species of Laelias from México.

This work is the first contribution on the relationship that exists between the thickness of the cuticle and the environment in the Mexican Laelias species. Particularly the species of the genus *Laelia* in México, which inhabit more xeric environments, have thicker cuticles as a mechanism of tolerance to drought, as has been reported for other orchid species belonging to xeromorphic environments [14].

In general, for the species of the Laeliinae subtribe, the shape of the cells in the epidermis is polygonal isodiametric, and their distribution is in rows [13]. In the case of Mexican Laelias, it was found that they present adaxial and abaxial epidermal cells, generally of an isodiametric polygonal shape; however, *L*. *eyermaniana*, *L*. *rubescens* and *L*. *superbiens* have elongated polygonal cells, and adaxial and abaxial epidermal cells are distributed alternately. In species such as *L*. *gouldiana*, *L*. *rubescens* and *L*. *halbingeriana,* the distribution is in rows.

One of the characters that is not considered in previous studies but quantitatively analyzed in our study was the area of the abaxial and adaxial epidermal cells. In the Mexican Laelias, it was found that the area of the epidermal cells is a character that helps differentiate the Mexican species of the genus *Laelia* from other species. For example, in the case of *Cattleya purpurata*, the area of the adaxial epidermal cells is >5000 µm^2^. In Mexican Laelias, the adaxial epidermal cell area is <5000 µm^2^, while abaxial cell area is 801 to 1400 µm^2^, except for *L. aurea*, where it is 600 to 800 µm^2^. The area of the abaxial epidermal cells in both species of the genus *Cattleya* is smaller than in all the Mexican Laelias species with a value of 600 to 800 µm^2^.

For the Mexican Laelias species, the thickness of the mesophyll was quantitatively measured, and it was found to range from 500 to 2500 µm, with *L*. *superbiens* being the species in this study with the greatest thickness. Furthermore, the type of mesophyll is also heterogeneous, except for *L*. *autumnalis*, *L*. *gouldiana*, *L*. *rubescens* and *L*. *superbiens*, where it is homogeneous.

*Laelia aurea* and *L. rubescens* are naturally exposed to a water deficit and are the only species with cells with secondary thickenings in the mesophyll. Baker reports the presence of secondary thickenings in the mesophyll as a common character in several species within the Laeliinae subtribe [13]. Pridgeon mentions that the presence of these cells in the mesophyll can function as additional water reservoirs [15,16]. It has been reported that these cells are present in several species of epiphytic orchids of the Oncidiinae subtribe, particularly those that inhabit hot and dry environments [17]. It is worth mentioning that cells with secondary thickenings in the mesophyll are not a particular character to all the Mexican species of *Laelia*, since only two species display them; nevertheless, they were also found in the species of *Broughtonia* and *Cattleya*, confirming its presence in several species within the Laeliinae subtribe [13].

Regarding vascular tissue, it has been reported that vascular bundles confer efficient water transport to the plant, necessary to respond to short periods of water availability [18]. The Mexican Laelias generally present four types of vascular bundles, except for *L. crawshayana*, *L. halbingeriana* and *L. superbiens*, which have three types, and species from more xeric environments such as *L. rubescens* that have up to five types of vascular bundles. There are two vascular bundle strata in the lamina, except for *L. eyermaniana* and *L. superbiens* with one stratum and *L. rubescens* with three strata. Throughout the lamina there are 30 to 90 vascular bundles, except in *L. superbiens*, which has more than 90 vascular bundles.

It is noteworthy that in this work it was found that species from more xeric environments have a greater number of vascular bundles in the mesophyll, have shorter leaves and are wider. Such is the case of *L. rubescens,* which has oblong-elliptical leaves up to 18 cm long and 4 cm wide [7]. In addition, it is interesting that the species *L*. *rubescens* that inhabits lowlands and has a high tolerance to drought [7], exhibits the greatest thickness of the mesophyll as well as three layers of vascular bundles in the mesophyll. On the other hand, the *Laelia* species that live in the high mountains have fewer vascular bundles, more elongated and narrower leaves, as well as a less thick mesophyll in contrast to the low-altitude Mexican Laelias.

Plants have mineral inclusions in some of their cells, such as crystals of various kinds. In about 75% of angiosperms, calcium oxalate crystals are the most abundant and are significant in taxonomy [19]. In Mexican Laelias, calcium oxalate crystals are present, except in *Laelia albida, L*. *anceps, L*. *aurea, L*. *autumnalis* and *L*. *eyermaniana*. The shape of these crystals is raphides and druse crystals; however, they are scarce, and their location is in the center of the mesophyll (Figure 9). These crystals were reported by Baker for the Laeliinae [13]. Calcium oxalate crystals are also present in other groups of Orchidaceae and raphidia, where they tend to be very abundant in both vegetative and reproductive structures [20]. 

*Laelia halbingeriana* is a natural hybrid and differs from *L. superbiens* by morphological peculiarities such as shorter and thicker pseudobulbs and a darkly bilobed anther [21]. However, both species live at medium altitudes (600 to 2000 m asl) in semi-deciduous tropical forests [7]. In this work, relevant characters were found in the foliar anatomy shared by these species, such as isodiametric polygonal shape of adaxial and abaxial epidermal cells, hypodermic cells with a lignified wall and heterogeneous mesophyll.

Finally, *Laelia aurea* and *L. rubescens* are species that stand out as phylogenetically related species that also share a morphological and anatomical similarity but are distinguished by the intense yellow color of the flowers and the shape of the more ovate pseudobulbs of *L*. *aurea*. Both species also inhabit xeric environments, mainly deciduous, dry tropical forests [7]. In a more recent study, Peraza Flores and collaborators made a molecular phylogeny from seven regions of plastid DNA and the ITS region (5′ partial 18S, ITS1, 5.8S, ITS2, 3′ partial 26S) where they included the Mexican species of *Laelia* [22]. They propose that the *Laelia* species need to be separated into two main clades, the *Laelia* Lindl. clade and the clade *Schomburgkia* Lindl. *L*. *aurea* and *L*. *rubescens*, the only Mexican species that grow in lowland environments, are immersed in the *Schomburgkia* clade. It is convenient to expand the sampling for anatomical studies where species of *Schomburgkia* are included, which will allow better understanding of the similarity between these two genera.

In the group of the Mexican Laelias species, *L*. *aurea* and *L*. *rubescens* were recently recognized as sufficiently different from the rest of *Laelia* to deserve genus status, for which Archila and Szlachetko proposed the creation of the genus *Encabarcenia* Archila & Szlach [23]. Nevertheless, based on molecular phylogeny, Peraza Flores and collaborators placed them in *Schomburgkia* for sharing low-elevation environments and other morphological characters [22]. In the present work, we showed that these two species had several structural characters in common and were different from the rest of the Laelias; therefore, we support the proposal of the latter and emphasize the importance of extending anatomical studies to include species of *Schomburgkia*.

From the combined analysis of morphological and anatomical data we were able to show that *Laelia* species from Mexico shared structural characters from which they can be distinguished from *Broughtonia negrilensis*, and to a greater extent from *Cattleya mossiae* and *Cattleya purpurata*, providing taxonomically useful information. It is important to emphasize that the present work provides information on the functional anatomical characters for the identification and distinction of Mexican Laelias such as area of abaxial epidermal cells, thickness of the abaxial cuticle and thickness of the mesophyll. This work also showed that within the Laelias of Mexico, there was a correlation between certain structural characters and the altitudinal distribution of their species. Taken together, these results contribute towards a better understanding of orchid evolution in Mexico.

## 4. Materials and Methods

### 4.1. Species Analyzed

The species of the Laeliinae subtribe are listed in Table 3. The species included in this study are taxonomically well distinguished. Since the purpose is not to know the infraspecific variation, only one individual of each one was included.

### 4.2. Analysis of Anatomical and Morphological Characters

The anatomical characters (Appendix A) were analyzed from histological preparations of the dermal (epidermis), fundamental (ground parenchyma) and vascular tissue of the leaves of these species with the illumination techniques in a bright field, phase contrast and polarization in a photomicroscope (Axioscop, Carl Zeiss^®^). These preparations belong to the collection of the Research Support Laboratory of the IB-UNAM Botanical Garden and were collected by the AMO Herbarium located in Mexico City. Measurements of the anatomical parameters were obtained with the Image-J program. We selected characters important for identification of the subtribe Laeliinae [13]. Regarding the sampling of the leaves, 30 measurements of one specimen were taken for each species from the middle and basal part of the leaf blades. Stomatal index (SI) was determined with the equation proposed by Salisbury [24].

The morphological characters include vegetative traits related to rhizomes, pseudobulbs, leaf, plus reproductive characters such as arrangement of inflorescence, flowers, floral bracts, the season of flowering, the column, ovary, petals, pollinia, viscidium and rostellum (Appendix A), based on Halbinger and Soto [7].

### 4.3. Anatomical Description of Laelias from Mexico

The foliar anatomy of the Laelias from Mexico was described, including the qualitative and quantitative characters of the demic, fundamental and vascular tissues. Likewise, we presented the evaluation of quantitative anatomical characters of the group.

### 4.4. Preparation of Basic Data Matrix and Statistical Analysis

The values corresponding to the mean, standard deviation, and the values (minimum and maximum) of the 11 quantitative anatomical characters of the species in this work were obtained. A matrix was made with these 11 continuous quantitative anatomical characters, each with 30 repetitions (Appendix A) for each species. To identify which characters had the greatest weight in the separation of groups, a canonical discriminant analysis (CDA) was performed for all species.

Descriptive statistics were performed for the 11 continuous quantitative anatomical characters. Subsequently, a test of normality (Shapiro–Wilk) and homoscedasticity of the variances (Levene’s test) were performed. A one-way analysis of variance (ANOVA) was applied to each character with 15 species as levels. These analyses and box plots for each character were prepared with R-Studio 3.6.2 [25].

In total, 107 morphological and anatomical characters were examined. The data matrix was elaborated from these structural characters to know the similarity relationships between the taxa from a cluster analysis. The character states of the quantitative anatomical characters were coded from a post hoc analysis with Dunnett’s test. The cluster analysis was developed with the total of the structural characters from the NTSYS-PC program (v.2.1), for which a mixed data matrix was generated including the quantitative characters and the qualitative characters encoded as binary or multistate. On other hand, a similarity matrix, obtained through the taxonomic distance coefficient and the average linkage (UPGMA), was used as the grouping algorithm of the NTSYS-PC program (v.2.1).

## Figures and Tables

**Figure 1 plants-12-01089-f001:**
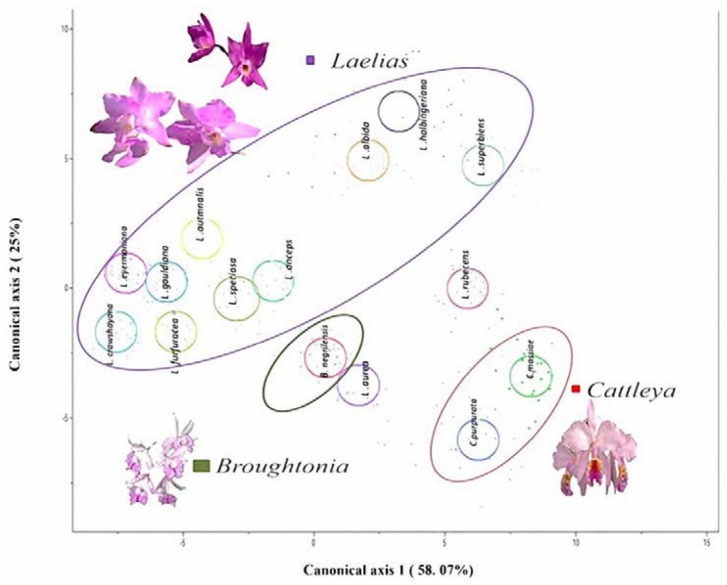
Canonical discriminant analysis (CDA) for the 15 species studied. Scatter diagram of points in the two-dimensional space of the CDA with 11 quantitative anatomical characters evaluated by species (canonical axis 1: 58.07%, canonical axis 2: 25%).

**Figure 2 plants-12-01089-f002:**
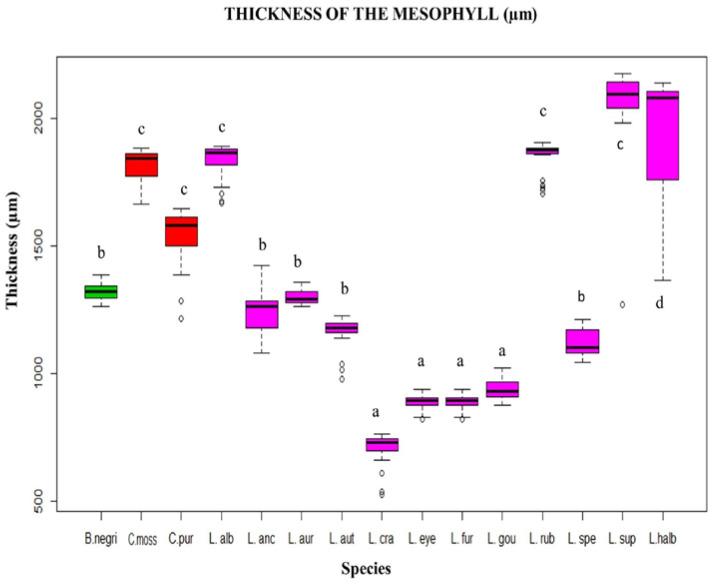
Box plot of mesophyll thickness evaluated by species. Each color identifies the genera: *Broughtonia* (green), *Cattleya* (red) and *Laelia* (pink). Each letter represents a group, according to Dunnett’s post hoc analysis. a = (500–1000 µm), b = (1001–1500 µm), c = (1501–2000 µm), d = (2001–2500 µm).

**Figure 3 plants-12-01089-f003:**
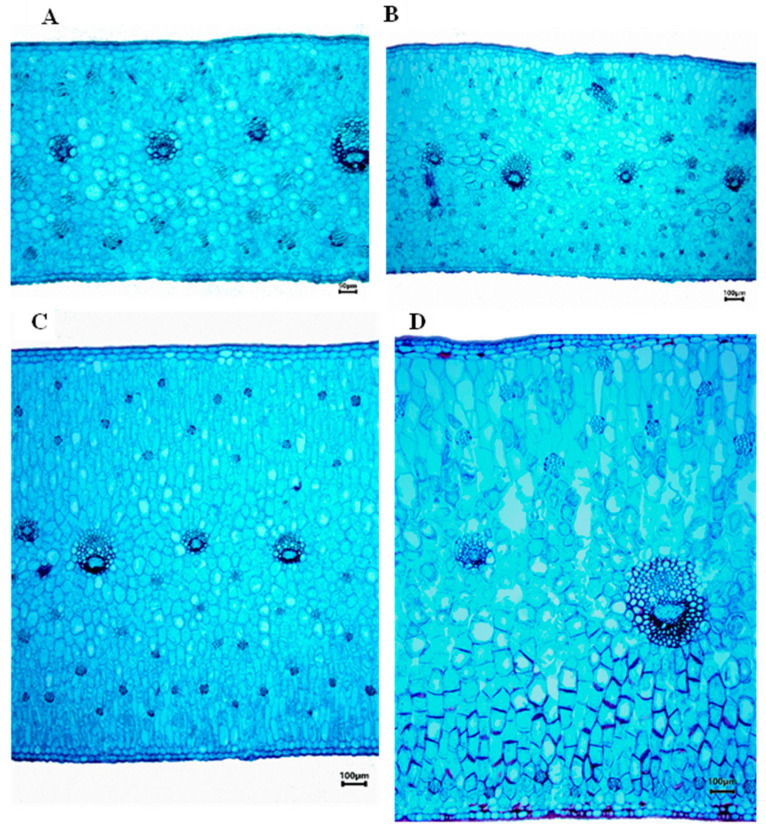
Transversal section (middle region). Mesophyll. (**A**) *Laelia crawshayana*, 500–1000 µm thick. (**B**) *Laelia furfuracea*: 1001 to 1500 µm. (**C**) *Laelia albida*: 1501 to 2000 µm. (**D**) *Laelia superbiens*: more than 2000 µm.

**Figure 4 plants-12-01089-f004:**
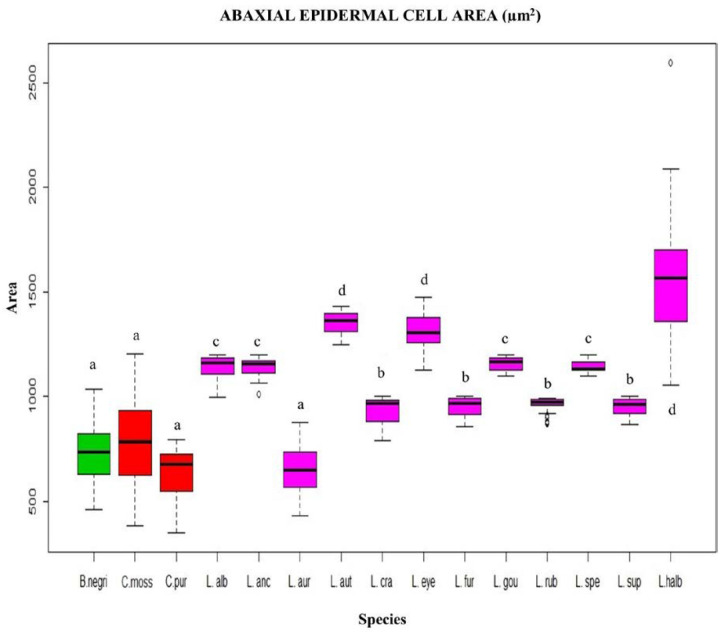
Box plot of the abaxial epidermal cell area, evaluated by species. Each color identifies the genera: *Broughtonia* (green), *Cattleya* (red) and *Laelia* (pink). Each letter represents a group, according to Dunnett’s post hoc analysis. a = 600 to 800 µm^2^, b = 801 to 1000 µm^2^, c = 1001 to 1200 µm^2^, d = 1201 to 1400 µm^2^.

**Figure 5 plants-12-01089-f005:**
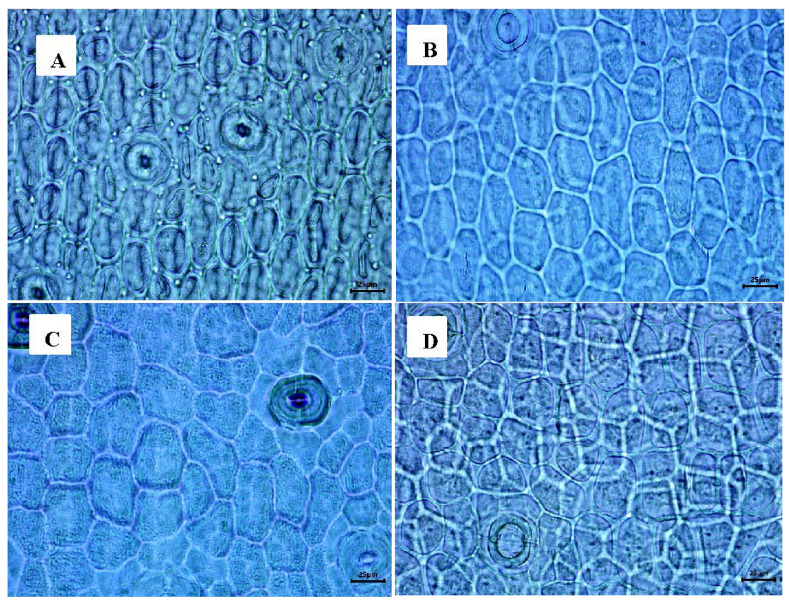
Area of abaxial epidermal cells. (**A**) *Cattleya mossiae* 600 to 800 µm^2^. (**B**) *Laelia crawshayana* 801 to 1000 µm^2^. (**C**) *Laelia anceps* 1001 to 1200 µm^2^. (**D**) *Laelia autumnalis* 1201 to 1400 µm^2^.

**Figure 6 plants-12-01089-f006:**
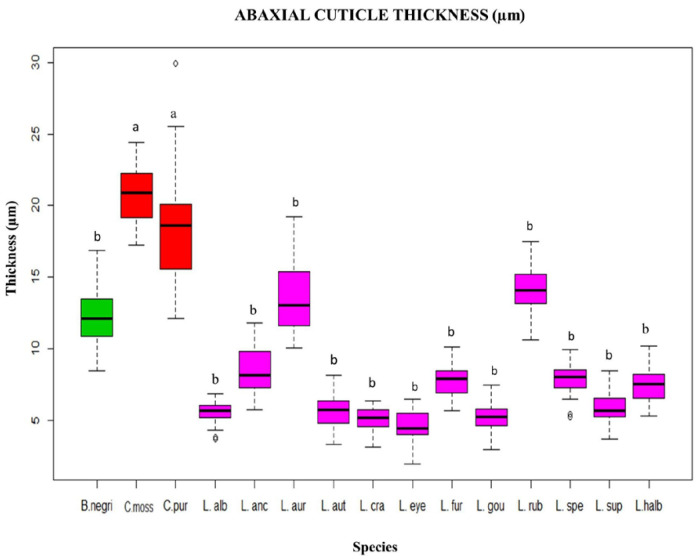
Box plot of the abaxial cuticle thickness evaluated by species. Each color identifies the genera: *Broughtonia* (green), *Cattleya* (red) and *Laelia* (pink). Each letter represents a group, according to Dunnett’s post hoc analysis. a = 0: ≥18 µm. b = 1: ≤18 µm.

**Figure 7 plants-12-01089-f007:**
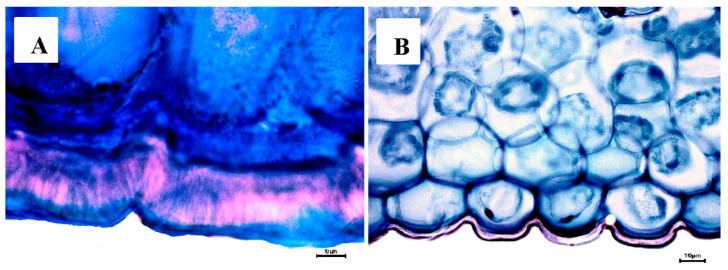
Abaxial cuticle thickness. (**A**) *Cattleya mossiae* (≥18 µm). (**B**) *Laelia crawshayana* (≤18 µm).

**Figure 8 plants-12-01089-f008:**
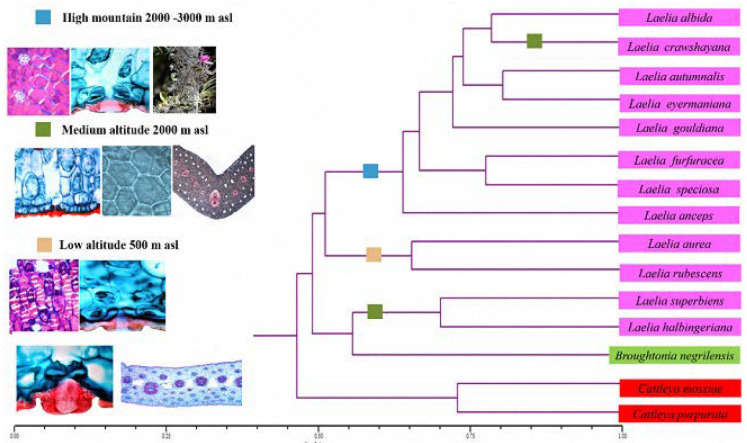
Phenogram of the anatomical and morphological characters with 15 species evaluating the taxonomic distance coefficient and the average linkage grouping algorithm (UPGMA). Note: the boxes represent an altitudinal distribution interval: blue (high mountain 2000–3000 m asl), green (medium altitude 2000 m asl), light brown (low altitude 500 m asl). The colors in the species represent the same genus: *Laelia* (pink), *Broughtonia* (green) and *Cattleya* (red).

**Figure 9 plants-12-01089-f009:**
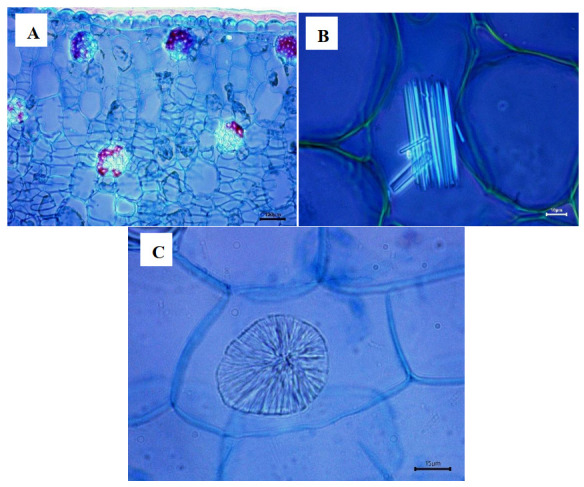
Presence of secondary thickenings and calcium oxalate crystals in the mesophyll. (**A**) *Laelia aurea* (cells with secondary thickening). (**B**) *Laelia superbiens*, crystals in the shape of raphides present in the mesophyll. (**C**) *Laelia rubescens,* druse crystals in the mesophyll.

**Table 1 plants-12-01089-t001:** Descriptive statistical data of the continuous quantitative characters of the 15 species analyzed in this study: *Laelia* in Mexico, *Broughtonia* and *Cattleya*. Numbers in bold indicate means, ± one standard deviation (SD); each letter represents a group, according to Dunnett’s post hoc analysis. Values indicate the mean, standard deviation, and (minimum and maximum) indicate the interval.

Variable	Group of *Laelia* in Mexico	*Broughtonia* (1 Specie)	*Cattleya* (2 Species)
Stomatal Index	**4.84** ± 1.32 b (1.39–8.43)	**3.32** ± 0.54 b (2.23–4.65)	**7.6** ± 1.41 a (4–11.2)
Abaxial epidermal cell area (μm^2^)	**1105.54** ± 255.84 a, b, c, d (432.11–2596.67)	**727.91** ± 135.35 a (461.01–1032.22)	**712.91** ± 190.55 a (347.86–1203.02)
Adaxial epidermal cell area (μm^2^)	**1604.54** ± 800.77 b (657.4–13,679.30)	**1017.59** ± 240.91 b (590.75–1459.25)	**3287.31** ± 1996.68 a, b (747.92–8740)
Epidermal cell width abaxial (μm)	**32.57** ± 7.29 a, b, c, d (15.05–53.68)	**28.94** ± 6.76 a (15.56–43.52)	**33.56** ± 6.86 b, c (15.83–48.22)
Epidermal cell height abaxial (μm)	**24.02** ± 5.44 b (11.04–44.04)	**21.21** ± 3.21 b (14.54–27.09)	**38.29** ± 7.85 a (21.18–56.33)
Epidermal cell width adaxial (μm)	**29** ± 6.23 a, b, c (13.91–49.63)	**25.78** ± 3.92 a (16.53–32.74)	**23.83** ± 6.28 a, c (12–41.09)
Epidermal cell height adaxial (μm)	**23.88** ± 4.92 b, c (10.98–41.11)	**17.91** ± 3.77 c (11.54–24.91)	**28.18** ± 10.39 a, c (12.71–54.15)
Abaxial cuticle thickness (μm)	**7.65** ± 3.35 b (2.02–19.25)	**12.13** ± 2.15 b (8.5–16.9)	**19.71** ± 3.15 a (12.15–29.9)
Adaxial cuticle thickness (μm)	**7.33** ± 3.54 a, b (2.42–18.3)	**13.63** ± 3 a (7.08–18.06)	**13.73** ± 5.20 a (7.55–28.92)
Mesophyll thickness (μm)	**1328.13** ± 462.70 a, b, c, d (528.23–2172.13)	**1322.14** ± 8.25 b (1261.39–1445.29)	**1678** ± 164.66 c (1216.17–1881.19)
Cell length occlusives (μm)	**36.33** ± 5.24 a, b, c (23.72–54.37)	**35.17** ± 1.97 c (31.09–39.17)	**33.96** ± 4.13 c (22.86–42.14)

**Table 2 plants-12-01089-t002:** Summary of the canonical discriminant analysis of 11 quantitative anatomical characters of the evaluated species.

Canonical Function	Value	Percentage%	Accumulated Percentage%	Canonical Correlation
1	27.627	58.074	58.074	0.982
2	11.867	24.946	83.020	0.960
3	3.350	7.042	90.063	0.877

**Table 3 plants-12-01089-t003:** Species of Laeliinae subtribe analyzed. The altitude and habitat are described for each species (data taken from Halbinger and Soto, 1997) [7]. *Als*: above sea level. Collectors: GS = Gerardo Salazar; MAS = Miguel Ángel Soto. s/n: no number assigned.

Species	Altitude	Habitat	Collector and Number of Collection
*Broughtonia negrilensis* Fowlie.	2000 m asl	Western Jamaica	MAS-s/n
*Cattleya mossiae* Hook.	800–2000 m asl	Rain Forest.	MAS-s/n
*Cattleya purpurata* (Lindl. & Paxton) Van Den Berg.	500 m asl	Brazil in near coastal areas.	MAS-s/n
*Laelia albida* Bateman ex Lindl.	2000–3000 m asl	North of Mexico high mountains, oak and juniper forest.	MAS-7431
*Laelia anceps* Lindl.	2000 m asl	Warm oak forest	GS-2438
*Laelia aurea* (La Llave & Lex.) Lindl.	500 m asl	Dry tropical forest and near coastal areas.	MAS-s/n
*Laelia autumnalis* (La Llave & Lex.) Lindl.	2000–3000 m asl	Oak forest and high mountains.	MAS-2352
*Laelia crawshayana* Rchb. F.	2000 m asl	Humid oak forest	Lamas-2352
*Laelia eyermaniana* Rchb. F	2000–3000 m asl	Oak forest and rocks.	MAS-2048
*Laelia furfuracea* Lindl.	2000–3000 m asl	Oak and juniper forest.	MAS-3839
*Laelia gouldiana* Rchb. F	2000–3000 m asl	Grows on mesquite trees, in a semiarid climate.	MAS-s/n
*Laelia halbingeriana* Salazar y Soto Arenas.	2000 m asl	Oak forest.	GS-6695
*Laelia rubescens* Lindl.	500 m asl	Dry deciduous tropical forest and in near coastal areas.	GS-s/n
*Laelia speciosa* Schltr.	2000–3000 m asl	Oak deciduous stunted forest.	MAS-s/n
*Laelia superbiens* Lindl.	2000 m asl	Mountain rain forest.	MAS-s/n

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
