# Peer review of "Contribution of Morphoanatomic Characters to the Taxonomy of the Genus LAELIA (Orchidaceae) in Mexico and Their Implication in Environmental Adaptation"

_plants, 2023, doi:10.3390/plants12051089_

Round 1

Reviewer 1 Report

Dear Authors,

The paper of Castellanos-Ramírez titled  Contribution of morphoanatomic characters to the taxonomy of the genus Laelia (Orchidaceae) in Mexico and their implication in environmental adaptation described the analysis of morphoanatomic characters of the 12 Lealia species in Mexico and closely related Cattleya and Broughtonia ones. This type of paper (results) are very important in taxonomical identification and/or comparison between Laelia species, when it is not used the genetic discrimination of them (with barcodes). However, without presentation the more detailed results (average values or the range of analysed traits) it is hard to discuss about them.  

Below, I suggested a few most important issues for consideration by authors:

Introduction

- the aim and hypothesis should be presented clearer in this paper rather than the statement (page 2, line 53)…to understand the similarity relationship between the Mexican species…

In Table 1 it can be added the altitudes and specific habitats in which the taxa occur.

Materials and methods

- it will be better to shown the morphological and anatomical characters as the lists in Appendix, because it is hard to understand what traits were taken into analysis.

-how many specimens/individuals were collected in analysis from each species. If only one, it can be a problem in estimation of mean and range of value of each trait.

-Stomatal indices for species were not shown in the text (in table?).

- If authors examine the morphological (vegetative and generative) characters, why the flower morphology will not describe in this paper. In this point, authors need to explain what morphological traits indicate differences between species and discuss with the other studies. Moreover, authors performed 11 anatomical characters but shown results only three of them in the paper.

Results

- authors noted that there were a few anatomical characters shared between High Mountain Lealias (page 8, line 192) based on the cluster analysis, but it is difficult to deliberate about this due to lack of presented results in the text. The same problem is on the 8th page lines (203-209).

Discussion

In this paragraph, authors compared Mexican Laelias  but as I mentioned above there were not revealed the detailed results (the averages values, etc.) of all characters to see the important differences between Laelias and Cattleya and Broughtonia species. Therefore, it is really hard to comment on these morphoanatomic comparisons between species. I propose also to stress only functional morphoanatomic traits which distinguish these studied species; it helps for quickly and reliable identification of species.

Reviewer 2 Report

This manuscript combines micro-morphological characters in Mexican genus Laelia  which have potentially adaptive significance and taxonomy. This idea is very interesting and brings important knowledge on this genus.

Unfortunately, the manuscript has weak points:

1.       Material and methods are described insufficiently, without detailed information on plants, species, organs (flowers, leaves), sampling and number of measurements

2.       In consequence, this is difficult to refers to both Results and Discussion parts

3.       Language needs correction

4.       Manuscript is carelessly edited

Therefore I recommend to reject the manuscript in present form  

Author Response

The detaided response is included in the attached file.

Round 2

Reviewer 1 Report

Dear Authors,

The paper of Montserrat Castellanos-Ramírez and co-authors “Contribution of morphoanatomic characters to the taxonomy of the genus Laelia (Orchidaceae) in Mexico and their implication in environmental adaptation” is considerably improved, and in my opinion it can be published. All my questions and suggestions were considered by the authors.

Yours sincerely,

Reviewer 2 Report

I am satisfied with the responses to the reviewers' comments and suggestions, as well as the corrections made to the manuscript. Therefore, I recommend the manuscript in its present form for publication in Plants.